# Clinical Characteristics of Adult Functional Constipation Patients with Rectoanal Areflexia and Their Response to Biofeedback Therapy

**DOI:** 10.3390/diagnostics13020255

**Published:** 2023-01-10

**Authors:** Fei Li, Meifeng Wang, Syed Hameed Ali Shah, Ya Jiang, Lin Lin, Ting Yu, Yurong Tang

**Affiliations:** Department of Gastroenterology, The First Affiliated Hospital with Nanjing Medical University, 300 Guangzhou Road, Gu Lou District, Nanjing 210029, China

**Keywords:** rectoanal areflexia, functional constipation, anorectal manometry, biofeedback

## Abstract

Background: The London Classification for anorectal physiological dysfunction specifically proposes rectoanal areflexia (RA), which means the absence of a rectoanal inhibitory reflex (RAIR) based on a manometric diagnosis. Although RA is not observed in healthy people, it can be found in adult patients with functional constipation (FC). This study describes the clinical manifestations of adult patients with FC and RA and their response to biofeedback therapy (BFT). Methods: This retrospective study reviewed the reports of high-resolution anorectal manometry (HR-ARM) and the efficacy of BFT in adult patients with FC. In addition, the Constipation Scoring System (CSS) scale, Patient Assessment of Constipation Symptoms (PAC-SYM) scale, Patient Assessment of Constipation Quality of Life (PAC-QOL) scale, Zung’s Self-Rating Anxiety Scale (SAS), Zung’s Self-Rating Depression Scale (SDS), balloon expulsion test (BET), and the use of laxatives were assessed. Results: A total of 257 adult patients diagnosed with FC were divided into the RA group (n = 89) and the RAIR group (n = 168). In the RA and RAIR groups, 60 (67.4%) and 117 (69.6%) patients, respectively, had dyssynergic defecation (DD) during simulated defecation. Type II pattern of dyssynergia was most frequently observed in both groups. Compared with the RA group, the RAIR group showed a higher CSS score, physical discomfort score, and prevalence of inadequate relaxation of the anal sphincter (*p* < 0.001, *p* = 0.036, and *p* = 0.017, respectively). The anxiety and depression scores were not different between the two groups. The proportion of patients using volumetric and stimulant laxatives and their combination was significantly higher in FC patients with RA, whereas the efficacy of BFT was significantly lower (*p =* 0.005, *p* < 0.001, *p* = 0.045, and *p* = 0.010, respectively). Conclusion: Adult FC patients with RA may suffer more severe constipation and have a lower efficacy of BFT compared with those with RAIR.

## 1. Introduction

Functional constipation (FC) is one of the most frequently occurring functional bowel disorders [1]. According to the latest research by the Foundation of Rome, the prevalence of FC in adults is 6.6–11.7% worldwide and around 6.2–10.6% in the Chinese population [1]. The clinical manifestations of FC include chronic difficulty in defecation, infrequent defecation, hard stools, incomplete defecation, and other gastrointestinal symptoms [2]. Psychological, behavioral, and neurological disorders are the etiologic factors associated with constipation [3,4]. Patients with FC experience a poor quality of life (QoL) paired with a heavy social burden [5,6].

High-resolution anorectal manometry (HR-ARM) is an important diagnostic tool for evaluating FC [7]. It assesses the presence of rectoanal inhibitory reflex (RAIR), the anal sphincter function, defecation dynamics, and rectal sensation [8]. RAIR, a physiological reflex mediated by an intramural nerve plexus in the anorectal area, helps in evacuation procedures and is the initiating factor for defecation [9]. Any morphological changes in the RAIR component can lead to anorectal dysfunction and intestinal constipation [10]. The London Classification for rectoanal physiological dysfunction based on objective measurements from the HR-ARM specifically proposes the RAIR disorder. The absence of a RAIR is termed rectoanal areflexia (RA) [8]. RA is a pattern that is not observed in healthy individuals but may be found in asymptomatic patients after rectal or anal surgery, anal hypotonia, stool impaction, or megarectum [11]. Both Hirschsprung’s disease (HD) and internal anal sphincter achalasia (IASA) show RA, which is common in children with constipation but rare in adult patients [12,13]. We found that RA is not rarely observed on HR-ARM in adult patients with FC. A recent study reported that children with IASA showed more severe constipation and usually do not respond to laxatives [14]. Most of these studies have focused on children with constipation, and few studies focused on RA in adult patients with FC [15,16]. At present, no specific study has explored the clinical characteristics and treatment of adult patients with FC and RA. Consequently, we aimed to investigate the clinical characteristics and the response to biofeedback therapy (BFT) in adult FC patients with RA (RA group) compared to those with RAIR (RAIR group).

## 2. Materials and Method

### 2.1. Study Design and Participants

This was a retrospective study conducted in the gastrointestinal motility clinic of the First Affiliated Hospital with Nanjing Medical University. Overall, 764 solid-state HR-ARM reports and balloon expulsion test (BET) results were reviewed from January 2016 to December 2020. For data accuracy, each HR-ARM report was double-checked by an anorectal manometry technician and a gastroenterologist. In addition, our gastrointestinal motility center has a professionally designed chronic disease management system and corresponding electronic medical information database. Clinical data were collected from every patient with chronic constipation (CC) at their first visit, and constipation-related evaluation was performed, including various relevant questionnaires. Patients were regularly followed up by telephone or face-to-face, especially after a certain course of therapy (e.g., five sessions of BFT). Periodic follow-up was carried out for patients who completed BFT five or more times to compare with the pre-treatment parameters to evaluate the efficacy of BFT and re-evaluate constipation after BFT. The follow-up data were recorded in the electronic database. In this retrospective study, patients’ clinical data including demographics, constipation symptoms, results of questionnaires, sessions of BFT, and imaging findings were extracted from the electronic medical information database and rechecked. The study was approved by the Ethics Committee of the hospital where the study was conducted (2021-SR-246).

### 2.2. Inclusion and Exclusion Criteria

The inclusion criteria were as follows: patients aged ≥ 18 years, those diagnosed with FC using the Rome IV criteria [17], those who completed the questionnaires covering the Constipation Scoring System (CSS) scale, the Patient Assessment of Constipation Symptoms (PAC-SYM) scale, the Patient Assessment of Constipation Quality of Life (PAC-QOL) scale, the Zung’s Self-Rating Anxiety Scale (SAS), the Zung’s Self-Rating Depression Scale (SDS), and received at least five sessions of BFT. Pregnant women, patients with a known diagnosis of any structural disease (such as a tumor, rectal prolapse, or intussusception) by colonoscopy, barium enema, or computed tomography (CT) scan, those with any surgery history of gastrointestinal or pelvic floor disorders, those with underlying chronic diseases (such as endocrine, metabolic, and neurological diseases, diagnosed with anxiety, or depression), those receiving medications such as hypnotic drugs or drugs that may affect bowel movements (such as antidepressants, spasmolytics, or opioids), and those with the presence of mental illness or cognitive impairment were excluded from this study.

### 2.3. Constipation Symptoms and Severity

The frequency of common constipation symptoms in patients with FC according to the Rome IV criteria [17] including infrequent defecation (<3 spontaneous bowel movements per week), hard stool (1–2 on the Bristol scale and appearing in >25% of defecations), excessive straining, sensation of incomplete evacuation, blockage, manual maneuvers to facilitate defecations (>25% of defecations), abdominal pain, and abdominal distension was assessed first.

The severity of constipation was determined by the modified CSS scale [18]. The scale consisted of eight items where a higher score represented more severe constipation. The PAC-SYM scale [19] aided in assessing each patient’s subjective feelings toward the perception of constipation. It contained 12 items, which were divided into three parts: defecation symptoms (five items), rectal symptoms (three items), and abdominal symptoms (four items). A higher total score illustrated a higher symptom burden.

### 2.4. Assessment of Mental Health

The SAS [20] and SDS [21] scores determined the level of anxiety and depression in each patient. Every questionnaire had 20 items that measured subjective feelings and the severity of anxiety or depression. Higher scores suggested severe psychological symptoms in the patients. In the Chinese population, a SAS score of ≥50 and SDS score of ≥53 indicated diagnosable anxiety and depression [22].

### 2.5. Patient Assessment of Constipation Quality of Life

The PAC-QOL scale evaluated the QoL of patients with constipation, which consisted of 28 items divided into four subscales (physical discomfort, psychosocial discomfort, worry/anxiety, and satisfaction with treatment) [23]. A higher total and subscale score indicated a poor QoL related to constipation.

### 2.6. High-Resolution Anorectal Manometry

Before the HR-ARM, each patient emptied their stool using glycerin enema and then a digital rectal examination was performed by a gastroenterologist to exclude stool impaction and local pathology. Before the HR-ARM, colonoscopy and abdominal CT scan were performed in all patients, and about 75.4% of the patients were subjected to a barium enema or defecography to exclude organic colorectal lesions such as the megarectum, colorectal surgery, and rectal prolapse. Anorectal manometric evaluation and rectal balloon distension were performed using a 12-sensor solid-state catheter with an expandable polyethylene balloon attached to the tip (ManoScan™ AR manometry system, Medtronic). The anorectal catheter was placed proximally at almost three cm to the superior edge of the anal sphincter with patients in the left lateral decubitus position. After five minutes of rest, the anal sphincter resting pressure (20–30 s) was evaluated followed by squeezing the anus (three times for the duration of 20–30 s). Simulated defecation (three attempts) was performed by asking the patient to bear down straining as if initiating a bowel movement. Along with these, the rectoanal pressure gradient (“rectal defecation pressure” minus “anal residual pressure” during the simulation of defecation) and the anal sphincter relaxation rate (ratio of “anal sphincter relaxation pressure” to “anal sphincter resting pressure”) were also evaluated. Inadequate relaxation of anal sphincter was identified when the anal sphincter relaxation pressure was ≤ 20% of anal sphincter resting pressure during the attempts of defecation. The absence of pressure reduction or an increase in the residual anal pressure during simulated defecation was defined as dyssynergic defecation (DD) [8,11]. DD was classified into four types according to the London Classification for anorectal disorders and the classification of defecation disorders described by Rao et al.: rectal pressure ≥ 40 mmHg along with paradoxical anal contraction (type I), rectal pressure < 40 mmHg along with paradoxical anal contraction (type II), rectal pressure ≥ 40 mmHg together with an absent or inadequate anal relaxation (type III), and rectal pressure < 40 mmHg together with an absent or inadequate anal relaxation (type IV) [8,11]. 

After stabilization of the anal sphincter basal pressure, RAIR was detected by rapidly inflating air into the intrarectal balloon in 10-mL increments (10–50 mL), and the distension threshold that caused the anal pressure decrease was measured, with the balloon deflated and a 30-second recovery interval between each inflating [8]. RAIR was considered normally present when the anal pressure decreased by >25% and absent when no RAIR appeared even after inflating 50 mL air in the balloon [24]. To ensure the accuracy of RAIR measurement, we repositioned the catheter and repeated the process twice in 50 mL if RAIR was not elicited at the first measurement [25,26]. The HR-ARM plots are represented in Figure 1. The rectal sensation was evaluated by gradually and uniformly distending the rectal balloon. For the first sensation, urgency, and maximum discomfort, the threshold volumes were recorded. Manometric data were analyzed and illustrated as pressure topography using the proprietary ManoView™ software. All pressure measurements were referenced to atmospheric pressure.

### 2.7. Balloon Expulsion Test

The time taken for expelling a rectal balloon filled with 50 mL of warm water while sitting on a commode was measured. An abnormal BET was described as the inability to expel the balloon in one minute or less [8,11]. 

### 2.8. Biofeedback Therapy

BFT was performed using the Polygraf ID8 biofeedback training system (Medtronic Ltd., Minneapolis, MN, USA. For each training session, the patients were asked to be in a right decubitus position and were covered with a sheet to simulate defecation. Three electrodes were attached to the lower abdomen, and an acryl plug was inserted into the patients’ anal canal at the anal sphincter. The acryl plug and the electrodes were connected to the Polygraf ID, which displays the changes in pressure activity in a simple graphical format. The signals associated with contracting and relaxing the pelvic floor muscles were observed on the computer monitor. The patients were instructed to increase intra-abdominal pressure and to contract and relax anus according to the prompts of the instrument. They can learn to regulate the activity of abdominal and anorectal muscles by repeating the simulated defecation training. Patients visited the gastrointestinal motility center for training three times per week, 60 minutes each time, 5–10 sessions for each course of BFT. During the therapy period, they were required to perform defecation force and relaxation training at home 2–3 times every day, 20 min each time [27].

### 2.9. Use of Laxatives and Assessment of Biofeedback Therapy Efficacy

In this study, four types of laxatives (volumetric, stimulant, osmotic, and lubricating) used after BFT were investigated. When patients completed at least five sessions of BFT, they were evaluated for their response to BFT using the CSS scale. The clinical outcome was recorded as “efficacious” if the score was ≥0.25, whereas it was marked as “no efficacy” if the score was <0.25. We compared the response rate to BFT in patients with FC in the two groups. Additionally, the difference in the BFT efficacy of patients with DD in the two groups was compared.
(1)$$Valid\;Score=\frac{Pre-training\;CSS\;score\;─\;Post-training\;CSS\;score}{Pre-training\;CSS\;score}$$

### 2.10. Statistical Analyses

Propensity score matching (PSM) was used to decrease confounders and balance the baseline variables between the RA and RAIR groups, using the 1–2 nearest neighbor method without replacement and a caliper width of 0.1 [28]. Matching was performed using variables that may affect the constipation assessment including age, gender, constipation course, and body mass index (BMI).

The statistical analyses were conducted using the SPSS for Windows (Version 26.0, IBM Corp., Armonk, NY, USA). Continuous variables were presented as mean ± standard deviation (SD) or median and interquartile ranges (IQR), whereas categorical variables were presented as numbers and percentages. To compare continuous variables, the independent samples *t*-test or the Mann–Whitney U test was used, whereas the chi-square test assessed the categorical variables. The results yielding *p* < 0.05 were considered statistically significant.

## 3. Results

### 3.1. Demographic Characteristics

Of the 764 HR-ARM reports reviewed, 666 patients were diagnosed with FC; among them, 175 patients had RA. The prevalence of RA in adult patients with FC by the HR-ARM test was, therefore, 26.3% in our institution. A total of 450 patients received ≥ 5 sessions of BFT and completed the posttreatment CSS questionnaire, of them, 92 patients had RA, and 358 had RAIR. After the PSM, 89 FC patients with RA and 168 FC patients with RAIR were included in the study (Figure 2). There was no significant variation in age, gender, constipation course, and BMI at baseline among the patients with RA and RAIR (Table 1).

### 3.2. Constipation Symptoms and Severity

The frequency of abdominal distension, excessive straining, the sensation of incomplete evacuation, the sensation of blockage, and infrequent bowel movements in the RA group was significantly higher than that in the RAIR group (*p* = 0.032, *p* = 0.012, *p* = 0.039, *p* = 0.008, and *p* = 0.001, respectively). A statistically insignificant difference was observed between the two groups in terms of abdominal pain, stool consistency, and manual maneuvers to facilitate defecation (*p* > 0.05; Figure 3). The CSS score of the patients in the RA group was significantly higher than that in the RAIR group (median [IQR]: 12.00 [9.50–14.50] vs. 9.50 [7.00–13.00], *p* < 0.001). However, no significant difference was observed in the PAC-SYM total score and subscale scores between the two groups (Table 2).

### 3.3. Assessment of Mental Health

The mean anxiety score of the RAIR and RA groups was 36.90 ± 12.38 and 34.31 ± 10.40, respectively, and the depression score was 50.11 ± 13.38 and 50.76 ± 10.67, respectively. The anxiety and depression scores demonstrated no statistically significant difference between the two groups (Table 2). However, the scores were higher than that of Chinese norms (anxiety score: 33.80 ± 5.90 and depression score: 41.88 ± 10.57). The incidence of depression was found to be 51.2% and 51.7% in patients of the RAIR and RA groups, respectively.

### 3.4. Assessment of Quality of Life

The “physical discomfort” score of the PAC-QOL scale between patients of the RAIR and RA groups showed a significant difference (*p* = 0.036), whereas there was an insignificant difference in the PAC-QOL total score and other subscale scores (*p* > 0.05; Table 2).

### 3.5. HR-ARM Metrics and BET Result

Of all the manometric parameters measured, the ratio of inadequate relaxation of the anal sphincter was significantly higher in patients of the RA group versus those in the RAIR group (57.3% [51/89] vs. 41.7% [70/168]; *p* = 0.017), whereas all other parameters were similar in both groups without a statistical significance (Table 3). There were few patients with a low baseline resting anal sphincter pressure, including 2 of 89 patients in the RA group and 5 of 168 patients in the RAIR group.

In total, 60 (67.4%) and 117 (69.6%) patients of the RA and RAIR groups, respectively, showed DD during simulated defecation with no statistically significant difference in the prevalence of DD (*p* = 0.714). Type II pattern of dyssynergia was most frequently observed in both two groups, accounting for 55% in the RA group and 50.4% in the RAIR group. Abnormal BET was observed in 64 (71.9%) patients of the RA group and 125 (74.4%) patients of the RAIR group with no significant difference in the proportion of abnormal BET (*p* = 0.666).

### 3.6. Use of Laxatives and BFT Efficacy

In total, 52 (58.4%) and 106 (63.1%) patients in the RA and RAIR groups, respectively, used any laxative but with no statistical significance (*p* = 0.464). A significant difference was observed in the proportion of volumetric laxatives (*p* = 0.005), stimulant laxatives (*p* < 0.001), and a combination of any two laxatives (*p* = 0.045; Table 4) between the two groups. 

The response rate to BFT was significantly low in FC patients with RA compared to those with RAIR (62.9% vs. 78.0%, *p* = 0.010). Moreover, the proportion of DD patients with RA who had effective BFT was significantly lower compared to those with RAIR (58.2% vs. 84.4%, *p* < 0.001; Figure 4).

## 4. Discussion

To our knowledge, this is the first study that has investigated the clinical characteristics and BFT efficacy in adult FC patients with RA compared to those with RAIR. The results demonstrated extensive and severe constipation symptoms in FC patients with RA compared to those with RAIR. Additionally, to date, no data have been reported on the prevalence of RA in adult FC patients. Among the study FC cohort, the prevalence of RA was 26.3%. Since our gastrointestinal motility center treats quite a high number of HR-ARM cases in China, we frequently come across patients with refractory constipation referred to us by other medical institutions. All patients were clinically assessed before HR-ARM and most of them showed indications of defecation disorder or mixed constipation. These factors may have contributed to the high prevalence of RA among our study cohort with FC. Though this observation needs to be studied in a larger Chinese population, our present results provide a close estimate based on the current data.

There are many possible causes of RA including megarectum, IASA, severely low anal resting pressure, post rectal resection or coloanal anastomosis, rectal ischemia, rectal prolapse, chronic constipation (CC), fecal incontinence, systemic sclerosis (SSc), diabetic neuropathy, myelomeningocele, and Chagas disease [9,10,12,25,29,30,31,32]. Examination-related reasons contributing to RA include displacements of catheters, artifacts, insufficient rectal balloon inflation, or stool impaction [14]. In our study, the organic causes for RA were excluded by reviewing detailed medical history, imaging examination, and other data when patients were enrolled. The influence of examination-related reasons was avoided as far as possible through the canonical HR-ARM process and strict review of test results. 

Previous studies have confirmed that the presence of RAIR depends on intramural pathways [9,14]. For example, RA in patients with HD is due to the congenital absence of ganglion cells in the colon or rectum, which prevents the reflexive relaxation of the internal anal sphincter (IAS) after rectal distention [30]. However, the pathophysiological mechanism of RA in non-neurogenic diseases such as FC is not fully known. One of the presumed mechanisms is persistent rectal distention in patients with CC due to prolonged retention of stool in the rectum that may lead to constant relaxation of the IAS and consequently the absence of RAIR [33]. V. A. Loening-Baucke reported that persistent rectal distention decreased the ability of the IAS to relax, which was observed in children with CC, which may be an underlying cause of constipation [34]. Another possibility is that patients with constipation may have an attenuation response to balloon distension due to impaired mechanoreceptor transduction across the rectal wall, which induces the absence of RAIR finally [33]. Some previous studies supported this hypothesis [15]. Further studies are required to verify these hypotheses. 

The most common symptom observed in our study was the sensation of incomplete evacuation, whereas the incidence of other constipation symptoms was >50% in both groups but was significantly higher in the RA group compared with the RAIR group. From the CSS scale, the constipation severity was significantly higher in the RA group than that in the RAIR group, but no obvious difference was observed in manometric parameters. The previous literature reported that chronic outlet obstructive constipation often occurs in patients with IASA or HD due to the absent relaxation of IAS [14,35], which is consistent with the results of our study.

No significant difference was found in the SAS and SDS scores between the two groups, although the scores were higher according to the Chinese norms, especially the SDS score, suggesting that depression may be more common in patients with FC. This is in accordance with a previous study [36]. Moezi et al. also showed a positive association between anxiety and depression with constipation [37]. 

Patients with FC have been demonstrated to experience poor QoL [38,39]. Johanson et al. reported that about 71% of patients with constipation considered that abdominal distention was affecting their overall QoL [40]. Our study indicated that FC patients with RA had severe constipation symptoms and worse “physical discomfort” subscale scores of PAC-QoL compared to patients with RAIR.

The anal sphincter relaxation rate and dyssynergia were detected during simulated defecation and primarily associated with the external anal sphincter (EAS) and pelvic floor muscles. However, the RA influences the contraction and relaxation of EAS and the opening of the anal canal, apparently due to an anatomical overlap between the IAS and EAS. Several studies reported that patients with CC have a lower percentage of IAS relaxation induced by rectal distention than healthy individuals [15,41] and a higher incidence of impaired or absent RAIR in patients with DD [11,16]. In the present study, the proportion of inadequate relaxation of the anal sphincter in the RA group was significantly higher than that in the RAIR group. The percentage of anal sphincter relaxation was slightly lower in the RA group compared with the RAIR group. Our results were in agreement with the previous studies, which suggested that FC patients with RA may be more likely to experience absent or inadequate anal sphincter relaxation.

No significant difference was observed in other dynamic parameters and the proportion of abnormal BET between the two groups. Patients in both groups showed similar HR-ARM sensation parameters, suggesting that the symptoms induced by RA may not be due to sensorial perception disorder but could be due to muscle malfunction, and these results were consistent with the previous literature [16,42]. 

In HR-ARM, 80% of the anal resting pressure is attributed to IAS and 20% to EAS [15]. RAIR helps in the reduction of intra-anal pressure and relaxation of IAS [43]. Low anal resting pressure may affect the detection of RAIR and is erroneously regarded as absent RAIR [8]. However, only two patients in the RA group and five patients in the RAIR group showed a low anal resting pressure according to our data, which had a negligible impact on the statistical results. We found that FC patients with RA showed a higher anal resting pressure, which may be related to the failure of IAS relaxation due to the absence of RAIR.

A study reported inadequate anorectal propulsion in 54.7% of the patients with FC [44]. We found that the rectal defecation pressure of the RA group was slightly lower than that of the RAIR group during simulated defecation and lower than the academic normal value (45 mmHg) in both groups, which was on par with the previous research. This may suggest inadequate defecatory propulsion is common in patients with FC and more serious in those of the RA group.

The mechanisms of DD are inadequate pushing force, paradoxical anal contraction, impaired anal relaxation, or a combination of these factors [11]. According to these features, it can be classified into four types [11]. Our data showed that type II dyssynergia was the most common, followed by type I dyssynergia in adult patients with FC, which was consistent with the previous reports [45]. The proportion of type II dyssynergia pattern in the RA group was slightly higher than that in the RAIR group, which may be related to the more serious inadequate rectal propulsion in the RA group as mentioned above. However, the proportion of type IV dyssynergia in both groups was very low, and further studies with a larger sample size may be needed.

The proportion of patients who used volumetric laxatives, stimulant laxatives, and a combination of any two laxatives was significantly more in the RA group compared with the RAIR group, which suggests that there was a higher need for and worse efficacy of laxatives in FC patients with RA. BFT has been widely used in the clinic, and the American Neurogastroenterology and Motility Society (ANMS) and the European Society of Neurogastroenterology and Motility (ESNM) consensus guidelines recommend BFT as the first-line treatment for DD [46]. As per the literature, the overall response rate to BFT in patients with FC was between 70% and 80% [47,48]. The overall efficacy of BFT was 80.7% in our institution, which is on par with the literature [27]. However, some patients with FC responded poorly to BFT. Predictive factors reported for the failure of BFT were CSS score severity before BFT and state of depression; older age, stool consistency, short duration of laxative use, and > 5 BFT sessions were known predictive factors for BFT success [27,48,49]. However, many predictors cannot be measured objectively. Our data showed that the response rate to BFT in FC and DD patients with RA was significantly lower than that in those with RAIR. These results may help clinicians to predict the efficacy of BFT and suggest that treatments other than BFT may be necessary for FC and DD patients with RA.

The lower response rate to BFT in FC patients with RA may lead to higher use of stimulant laxatives to improve their symptoms. Some studies that investigated the effects of botulinum toxin injections on the anal sphincter rendered it as a well-tolerated and safe intervention with a positive response in IASA and HD [12,13,50]; hence, it may be viable for patients with FC and RA who have a poor response to BFT.

The present study has some limitations. Firstly, possible selection bias due to the retrospective nature of the study. Secondly, additional medications or therapies may have influenced the response rate. This can be overcome by adequately controlling the medication in future studies. Our study results need to be confirmed in a prospective cohort, essentially, on the outcomes of FC patients with RA. 

## 5. Conclusions

In conclusion, RA is not rare in adult patients with FC, and adult FC patients with RA may have more severe constipation symptoms, a higher need for laxatives, and a lower efficacy of BFT than those patients with RAIR. Based on these findings, HR-ARM is recommended for adult patients with FC to detect RAIR, especially for those who need to receive BFT.

## Figures and Tables

**Figure 1 diagnostics-13-00255-f001:**
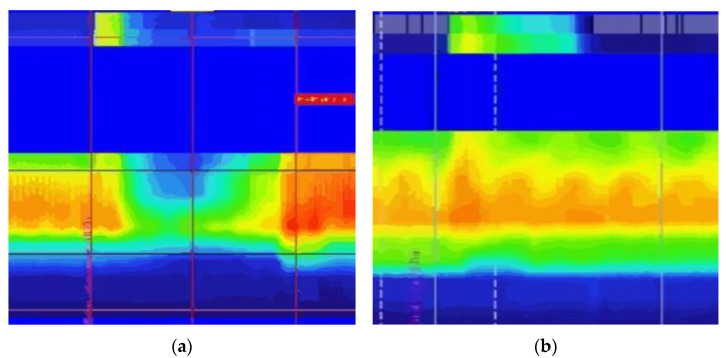
High-resolution anorectal manometry studies during balloon inflation. (**a**) Patients with rectoanal inhibitory reflex (RAIR) and (**b**) patients with rectoanal areflexia (RA). The upper part shows the pressure of the balloon inflation, and the lower part shows the effect on the anal canal. The pressure gradually displays from dark blue (lowest pressure = 0 mmHg) to red (highest pressure = 150 mmHg).

**Figure 2 diagnostics-13-00255-f002:**
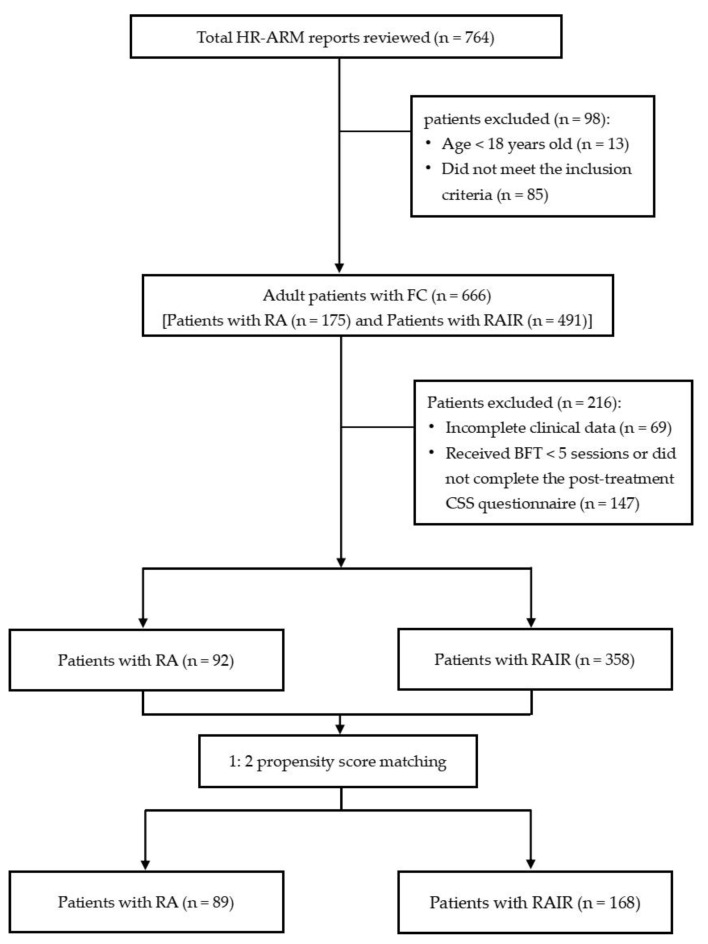
Flowchart for patient disposition. Abbreviations: BFT, biofeedback therapy; CSS, Constipation Scoring System; FC, functional constipation; HR-ARM, high-resolution anorectal manometry; RA, rectoanal areflexia; RAIR, rectoanal inhibitory reflex.

**Figure 3 diagnostics-13-00255-f003:**
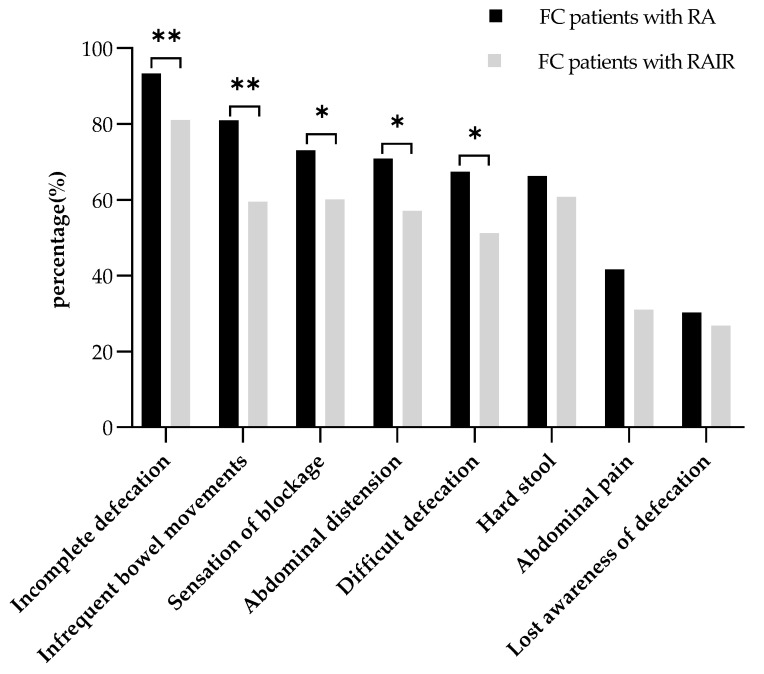
Frequency of common constipation symptoms in FC patients with RA and those with RAIR using the chi-squared test. * *p* < 0.05; ** *p* < 0.01. Abbreviations: FC, functional constipation; RA, rectoanal areflexia; RAIR, rectoanal inhibitory reflex.

**Figure 4 diagnostics-13-00255-f004:**
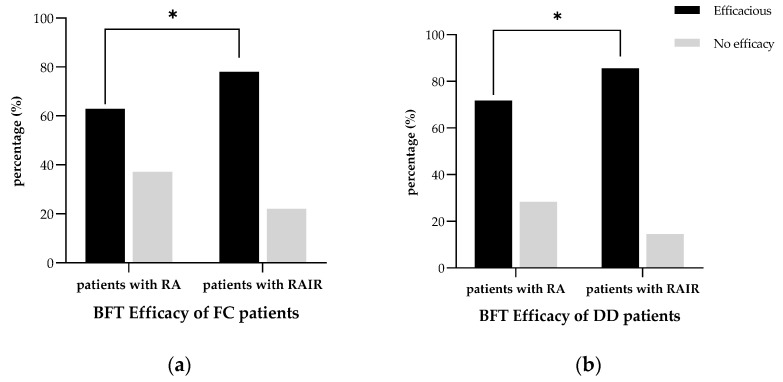
The response rate to BFT: (**a**) patients with FC and (**b**) patients with DD. * *p* < 0.05 using the chi-squared test. Abbreviations: BFT, biofeedback therapy; DD, dyssynergic defecation; FC, functional constipation; RA, rectoanal areflexia; RAIR, rectoanal inhibitory reflex.

**Table 1 diagnostics-13-00255-t001:** Baseline data after the PSM analysis.

Variables	RA Group (n = 89)	RAIR Group (n = 168)	*p*-Value
Age (years)	44.07 ± 14.58	44.32 ± 14.14	0.895
Female (n, %)	75 (84.3)	137 (81.5)	0.585
Constipation course (years)	6.00 (3.00–13.00)	5.70 (2.00–10.00)	0.257
BMI (kg/m^2^)	21.88 (19.59–23.47)	21.41 (19.57–24.22)	0.744

**Table 2 diagnostics-13-00255-t002:** The scores of different scales.

		RA Group (n = 89)	RAIR Group (n = 168)	*p*-Value
Constipation severity	CSS	12.00 (9.50–14.50)	9.50 (7.00–13.00)	**<0.001**
	PAC-SYM	1.50 (0.96–1.88)	1.38 (0.92–1.75)	0.163
	Abdominal	1.00 (0.50–1.50)	1.00 (0.25–1.50)	0.099
	Rectal	0.33 (0.00–1.00)	0.33 (0.00–1.00)	0.431
	Stool	3.67 (2.33–5.00)	3.67 (2.67–4.67)	0.517
Mental health	SAS	34.31 ± 10.40	36.90 ± 12.38	0.094
	SDS	50.76 ± 10.67	50.11 ± 13.38	0.692
Constipation-related quality of life	PAC-QOL	1.50 (0.95–2.04)	1.48 (0.96–2.07)	0.926
	PD	1.50 (0.88–2.00)	1.25 (0.75–1.75)	**0.036**
	PSD	1.00 (0.63–1.38)	0.88 (0.38–1.50)	0.470
	Worry/anxiety	1.55 (0.77–2.27)	1.64 (0.82–2.34)	0.667
	Satisfaction	2.60 (1.40–3.20)	2.60 (1.60–3.40)	0.615

CSS, Constipation Scoring System; PAC-SYM, Patient Assessment of Constipation Symptoms; PAC-QOL, Patient Assessment of Constipation Quality of Life; PD, physical discomfort; PSD, psychosocial discomfort; RA, rectoanal areflexia; RAIR, rectoanal inhibitory reflex; SAS, Zung’s Self-Rating Anxiety Scale; SDS, Zung’s Self-Rating Depression Scale.

**Table 3 diagnostics-13-00255-t003:** Manometric parameters.

HR-ARM Metrics	RA Group (n = 89)	RAIR Group (n = 168)	*p*-Value
Anal resting pressure (mmHg)	96.19 ± 25.62	91.44 ± 24.95	0.151
Low anal resting pressure (n, %)	2 (2.2)	5 (3.0)	0.733
Maximum squeeze pressure (mmHg)	236.72 ± 60.98	245.41 ± 70.09	0.324
Duration of sustained squeeze (seconds)	20.00 (16.20–21.45)	19.25 (13.73–20.70)	0.096
Rectal defecation pressure (mmHg)	35.96 ± 20.16	38.69 ± 19.82	0.297
Anal residual pressure (mmHg)	79.52 ± 31.55	83.57 ± 29.54	0.309
Rectoanal pressure gradient (mmHg)	−43.57 ± 34.79	−44.88 ± 35.43	0.776
Anal sphincter relaxation rate (%)	17.00 (10.00–40.00)	26.00 (10.25–41.00)	0.285
Inadequate relaxation of the anal sphincter (n, %)	51 (57.3)	70 (41.7)	0.017
First sensitive volume (mL)	40.00 (30.00–60.00)	40.00 (30.00–63.75)	0.402
Urge to defecate (mL)	90.00 (70.00–120.00)	100.00 (75.00–130.00)	0.165
Maximum tolerable volume (mL)	140.00 (100.00–180.00)	150.00 (100.00–200.00)	0.495
Dyssynergia (n, %)	60 (67.4)	117 (69.6)	0.714
Dyssynergia pattern			0.427
Type I (n, %)	22 (36.7)	38 (32.5)	
Type II (n, %)	33 (55.0)	59 (50.4)	
Type III (n, %)	3 (5.0)	9 (7.7)	
Type IV (n, %)	2 (3.3)	11 (9.4)	
Abnormal BET (n, %)	64 (71.9)	125 (74.4)	0.666

BET, balloon expulsion test; HR-ARM, high-resolution anorectal manometry; RA, rectoanal areflexia; RAIR, rectoanal inhibitory reflex.

**Table 4 diagnostics-13-00255-t004:** Use of laxatives.

Laxatives	RA Group (n = 89)	RAIR Group (n = 168)	*p*-Value
Use of total laxatives	52 (58.4)	106 (63.1)	0.464
Use of volumetric laxatives	49 (55.1)	62 (36.9)	0.005
Use of stimulant laxatives	42 (47.2)	30 (17.9)	<0.001
Use of osmotic laxatives	11 (12.4)	10 (6.0)	0.074
Use of lubricating laxatives	9 (10.1)	30 (17.9)	0.100
Combination of two laxatives	19 (21.3)	20 (11.9)	0.045

RA, rectoanal areflexia; RAIR, rectoanal inhibitory reflex.

## Data Availability

The authors declare that the data of this study are available from the corresponding author on reasonable request.

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
