# Peer review of "Clinical Characteristics of Adult Functional Constipation Patients with Rectoanal Areflexia and Their Response to Biofeedback Therapy"

_diagnostics, 2023, doi:10.3390/diagnostics13020255_

Round 1

Reviewer 1 Report

I think it is well comsidered.

Author Response

Point: I think it is well considered.

Response: Thank you very much for taking the precious time to review the manuscript and giving full affirmation to our manuscript.

Reviewer 2 Report

In this interesting submission, the authors provide the results of a retrospective study focusing on two subgroups of patients with functional constipation (with /without recto-anal areflexia).

The topic is of great interest, also in the light of the prevalence of this problem in the general population. I am pleased to say that the paper is methodologically sound and the results are presented in detail.

The major concern relates to its retrospective nature. In this context, I wonder how the authors could obtain all the information regarding the scores and QoL questionnaires. 

The second concern relates the effect of biofeedback treatment. How did you get pre/post-treatment information retrospectively. Additionally, how was biofeedback applied?

In summary, while the exploratory part of this study is of interest (i.e. the individuation of a subgroup of patient with difficult-to-treat constipation), the other conclusions seem not to be adequately supported. Therefore, I would recommend the author to review their submission at the light of these observation.

Additionally, a future prospective study would be more than welcome by the medical community.

Author Response

Response to Reviewer 2’s Comments

In this interesting submission, the authors provide the results of a retrospective study focusing on two subgroups of patients with functional constipation (with /without recto-anal areflexia).

The topic is of great interest, also in the light of the prevalence of this problem in the general population. I am pleased to say that the paper is methodologically sound and the results are presented in detail.

Response: Thank you very much for your time involved in reviewing the manuscript and your very encouraging comments on the merits.

Point 1: The major concern relates to its retrospective nature. In this context, I wonder how the authors could obtain all the information regarding the scores and QoL questionnaires.

Response 1: Thank you for your review. Our gastrointestinal motility center has a professionally designed chronic disease management system and corresponding electronic medical information database. We collect clinical data and conduct constipation-related evaluation for every patient with chronic constipation at the first visit, including various questionnaires such as PAC-QOL. Patients are regularly followed up by telephone or face to face, especially after a certain course of therapy (e.g., five sessions of BFT). The follow-up data were also entered into the electronic database. In this retrospective study, we extracted the data and checked it again. We have added detailed instructions to the part “2.1 Study Design and Participants” of the manuscript (Line71-83 ).

  • Change made: In addition, our gastrointestinal motility center has a professionally designed chronic disease management system and corresponding electronic medical information database. Clinical data were collected from every patient with chronic constipation (CC) at their first visit, and constipation-related evaluation was performed, including various relevant ques-tionnaires Patients were regularly followed up by telephone or face to face, especially after a certain course of therapy (e.g., five sessions of BFT). Periodic follow-up was done for patients who completed BFT for five or more times to compare with the pre-treatment parameters to evaluate the efficacy of BFT and re-evaluate the constipation after BFT. The follow-up data would also be entered into the electronic database. In this retrospective study, those patients’ clinical data including demographics, constipation symptoms, results of questionnaires, sessions of BFT, and imaging findings were extracted from the electronic medical information database and re-checked.

Point 2: The second concern relates the effect of biofeedback treatment. How did you get pre/post-treatment information retrospectively. Additionally, how was biofeedback applied?

Response 2: â‘  Thank you for the query. As previously mentioned, pre-treatment information was collected from every patient during the first visit. During BFT treatment, periodic follow-ups were done and information was collected from patients with chronic constipation to evaluate the efficacy of BFT and re-evaluate the constipation. Both the pre-treatment and post-treatment parameters were maintained on the electronic medical information database which was subsequently used for extraction of patient data. We have added detailed instructions in the part “2.1 Study Design and Participants” of the manuscript (Line71-83 ).

  • Change made: In addition, our gastrointestinal motility center has a professionally designed chronic disease management system and corresponding electronic medical information database. Clinical data were collected from every patient with chronic constipation (CC) at their first visit, and constipation-related evaluation was performed, including various relevant ques-tionnaires. Patients were regularly followed up by telephone or face to face, especially after a certain course of therapy (e.g., five sessions of BFT). Periodic follow-up was done for patients who completed BFT for five or more times to compare with the pre-treatment parameters to evaluate the efficacy of BFT and re-evaluate the constipation after BFT. The follow-up data would also be entered into the electronic database. In this retrospective study, those patients’ clinical data including demographics, constipation symptoms, results of questionnaires, sessions of BFT, and imaging findings were extracted from the electronic medical information database and re-checked.

â‘¡ Thank you for the great suggestion. We have added the biofeedback therapy protocol in the part “2.8 Biofeedback therapy” of the manuscript (Line 172-186 ).

This protocol is based on some previously published literature of our gastrointestinal motility center (Reference: Yu T, Shen X, et al. Efficacy and Predictors for Biofeedback Therapeutic Outcome in Patients with Dyssynergic Defecation. Gastroenterol Res Pract. 2017;2017:1019652. Epub 2017/09/28. doi: 10.1155/2017/1019652. )

  • Change made: 8. Biofeedback therapy

BFT was performed using the Polygraf ID8 biofeedback training system (Medtronic Ltd, Denmark). Each training session was performed and patients were asked to be in a right decubitus position and were covered with a sheet to simulate defecation. Three electrodes were attached to the lower abdomen and an acryl plug was inserted into the patient’s anal canal at the anal sphincter. The acryl plug and the electrodes were connected to the Polygraf ID, which displays the changes of pressure activity in a simple graphic format. The signals associated with contracting and relaxing of the pelvic floor muscles could be observed on the computer monitor. The patients were instructed to increase intra-abdominal pressure, and to contract and relax anus according the prompts of the instrument. They can learn to regulate the activity of abdominal and anorectal muscles by repeating the simulated defecation training. Patients visited the gastrointestinal motility center for training three times per week, 60 minutes each time, 5-10 sessions for each course of BFT. During the therapy period, they were required to perform defecation force and relaxation training at home two to three times every day, 20 minutes each time [27].

Point 3: In summary, while the exploratory part of this study is of interest (i.e. the individuation of a subgroup of patient with difficult-to-treat constipation), the other conclusions seem not to be adequately supported. Therefore, I would recommend the author to review their submission at the light of these observation. 

Response 3: Thank you for the comment.

  • Change made: According to your suggestion, we have carefully reviewed the conclusions of our manuscript and removed the phrase "poor QoL" from the conclusions (Line 402) and “worse quality of life” from the abstract (Line 33).
  • Detailed explanations as follows:

â‘  “In conclusion, the RA is not rare in adult FC patients”: In this study, we found that 26.3% of adult patients with FC have RA, close to 1/3. Therefore, we considered that RA is not rare in adult patients with FC. We still reserve this conclusion.

â‘¡ “and adult FC patients with RA may have more severe constipation symptoms, poor QoL, a higher need for laxatives, and lower efficacy of BFT than those patients with RAIR.” : In this study, we compared the PAC-QOL scores of FC patients in the RA and RAIR group, and found that only the “physical discomfort” category had a statistical difference (which may be related to sampling bias in the retrospective study). Indeed, it cannot be directly summarized as "the QoL of patients in the RA group was worse". We have removed the phrase "poor QoL" from the conclusions (Line 402) and “worse quality of life” from the abstract (Line 33). Our study found that the prevalence of constipation symptoms and CSS score of patients in the RA group were significantly higher than those in the RAIR group. In addition, the proportion of patients in the RA group who needed to use volumetric laxatives, stimulant laxatives and the combination of two laxatives was significantly higher than that in the RAIR group; while the efficacy of BFT in the RA group was significantly lower than that in the RAIR group, with statistical differences in these results. So, we considered that FC patients with RA have more severe constipation symptoms, a higher need for laxatives, and a poorer BFT outcome. We intend to keep this part of the conclusion.

Point 4: Additionally, a future prospective study would be more than welcome by the medical community.

Response 4: Thank you for suggesting this point.

We fully agree with you and have added this point to the limitations of this study (Line 395-399). We will conduct a future prospective study to further explore RA. Special thanks to you for your good comments.

  • Change made: The present study has some limitations: possible selection bias due to the retrospective nature of the study. Moreover, additional medications or therapies could be adequately controlled during the BFT process, which may have subsequent positive or negative effects on the response rate. Future prospective cohort study should be performed to confirm our results and focus on evaluating the outcomes of FC patients with RA.

Reviewer 3 Report

The authors aimed to investigate clinical characteristics and responses to the biofeedback treatment in adult FC patients with rectoanal areflexia (RA) vs. those with RAIR. Although it is a retrospective study, the sample size is large and valuable conclusions can be drawn with an appropriate study design. I have some comments that I would like the authors to clarify.

1. The maximum volume authors use as the cutoff point for determining RA is very important. This definition should be mentioned clearly in the methods. Otherwise, all the results cannot be interpreted correctly.
2. Rectal sensations were comparable between RA and RAIR groups, and most participants had no megaractum. Authors should discuss what is/are likely to be the cause of RA and should add the numbers of participants that have undergone radiological evaluation for rectal lesions.
3. Biofeedback therapy protocol should be added.

Author Response

Response to Reviewer 3’s Comments

The authors aimed to investigate clinical characteristics and responses to the biofeedback treatment in adult FC patients with rectoanal areflexia (RA) vs. those with RAIR. Although it is a retrospective study, the sample size is large and valuable conclusions can be drawn with an appropriate study design. I have some comments that I would like the authors to clarify.

Response: Thank you very much for taking your precious time to review the manuscript and giving your encouraging comments.

Point 1: The maximum volume authors use as the cutoff point for determining RA is very important. This definition should be mentioned clearly in the methods. Otherwise, all the results cannot be interpreted correctly.

Response 1: Thank you for the comment. In our study, the cutoff point for determining RA that we used was inflated with 50 mL air. If RAIR could not be induced with injecting 50 mL air and repeated the process twice in 50 mL, it was determined to be RA.

The protocol and cutoff point for RA were based on some previous literature. There is no unified value of the cut-off point for determining RA now, and the London Classification also  does not provide a definite cutoff value. Therefore, based upon several reports related to RA determination mentioned below, we have found that most of them used 50ml cutoff point, while there were few articles where 60ml cutoff point was used. Considering the factors such as ethnic differences, we chose the protocol of RA determination to follow the majority of the reported literature, especially the study of a Chinese multi-center study of normative values of the anorectal manometry.

References:

â‘  Sun Xiaohong WZ, et al. Normative values and its clinical significance of the anorectal manometry in Chinese from multi-center study. Chin J Dig. 2014;34:597-602. Epub 2014/09/05. doi: 10.3760/cma.j.issn.02541432.2014.09.005.

â‘¡ Thiruppathy K, Bajwa A, Kuan KG, Murray C, Cohen R, Emmanuel A. Gut symptoms in diabetics correlate with components of the rectoanal inhibitory reflex, but not with pudendal nerve motor latencies or systemic autonomic neuropathy. J Dig Dis. 2015;16(6):342-9. Epub 2015/03/13. doi: 10.1111/1751-2980.12244. PubMed PMID: 25761939.

â‘¢ Scott SM, Carrington EV. The London Classification: Improving Characterization and Classification of Anorectal Function with Anorectal Manometry. Curr Gastroenterol Rep. 2020;22(11):55. Epub 2020/09/17. doi: 10.1007/s11894-020-00793-z. PubMed PMID: 32935278.).

  • Change made: We have made more detailed revisions and explanations in the part “2.6 High-resolution anorectal manometry” of the manuscript (Line 155-157).

RAIR was considered normally present when the anal pressure decreases by > 25%, and absent when no RAIR appeared even after inflating 50 mL air in the balloon [24]. To ensure the accuracy of RAIR measurement, the catheter would be repositioned and the process would be repeated twice in 50 mL if RAIR was not elicited with 50 mL air [25, 26].

Point 2: Rectal sensations were comparable between RA and RAIR groups, and most participants had no megaractum. Authors should discuss what is/are likely to be the cause of RA and should add the numbers of participants that have undergone radiological evaluation for rectal lesions.

Response 2: Thank you for suggesting this point.

  • Change made: We have added the detailed discussion to the manuscript (Line 293-316).

There are many possible causes of RA including megarectum, IASA, severely low anal resting pressure, post rectal resection or coloanal anastomosis, rectal ischemia, rectal prolapse, chronic constipation (CC), fecal incontinence, systemic sclerosis (SSc), diabetic neuropathy, myelomeningocele, Chagas disease and so on [9, 10, 12, 25, 29-32]. Examination-related reasons responsible for RA include displacements of catheters, artifacts,  insufficient rectal balloon inflation, or stool impaction [14]. In our study, the organic causes for RA were excluded by reviewing detailed medical history, and imaging examination and other data when patients were enrolled, and the influence of examination-related rea-sons were avoided as far as possible through canonical HR-ARM process and strict re-view of test results.

Previous studies have confirmed that the presence of RAIR depends on intramu-ral pathways [9, 14]. For example, the RA in patients with HD due to the congenital absence of ganglion cells in the colon or rectum, which prevents the reflexive relaxa-tion of internal anal sphincter (IAS) after rectal distention [30]. However, the patho-physiological mechanism of RA in non-neurogenic diseases such as FC is not fully known. One of the presumed mechanisms is that persistent rectal distention in pa-tients with CC due to prolonged retention of stool in the rectum that may lead to constant relaxation of the IAS and consequently the absence of RAIR[33]. V. A. Loening-Baucke reported that the persistent rectal distention decreased the ability of the IAS to relax that was observed in children with CC, which may be a underlying cause of constipation [34]. Another possibility is that patients with constipation may have an attenuation response to balloon distension due to impaired mechanoreceptor transduc-tion across the rectal wall, which induces the absence of RAIR finally [33].  Some pre-vious studies supported this hypothesis [15]. These hypotheses may need to be further verified by future researches.

  • Change made: According to your suggestion, we have added the specific number and proportion of patients who received radiological evaluation for rectal lesions in the manuscript (Line 126-129).

Before the HR-ARM, all patients received the colonoscopy and abdominal CT scan, and about 75.4% of the patients got a barium enema or defecography to exclude organic colonrectal lesions such as the megarectum, colorectal surgery, rectal prolapse, and so on

Point 3: Biofeedback therapy protocol should be added.

Response 3: Thank you for the great suggestion and we strongly agree with your point of view. We have add the biofeedback therapy protocol in the part “2.8 Biofeedback therapy” of the manuscript (Line 172-186).

Additionally, this protocol is based on some published literature of our gastrointestinal motility center (Reference: Yu T, Shen X, et al. Efficacy and Predictors for Biofeedback Therapeutic Outcome in Patients with Dyssynergic Defecation. Gastroenterol Res Pract. 2017;2017:1019652. Epub 2017/09/28. doi: 10.1155/2017/1019652. )

  • Change made: 8. Biofeedback therapy

BFT was performed using the Polygraf ID8 biofeedback training system (Medtronic Ltd, Denmark). Each training session was performed and patients were asked to be in a right decubitus position and were covered with a sheet to simulate defecation. Three elec-trodes were attached to the lower abdomen and an acryl plug was inserted into the pa-tient’s anal canal at the anal sphincter. The acryl plug and the electrodes were connected to the Polygraf ID, which displays the changes of pressure activity in a simple graphic for-mat. The signals associated with contracting and relaxing of the pelvic floor muscles could be observed on the computer monitor. The patients were instructed to increase intra-abdominal pressure and to contract and relax anus according to the prompts of the instrument. They can learn to regulate the activity of abdominal and anorectal muscles by repeating the simulated defecation training. Patients visited the gastrointestinal motility center for training three times per week, 60 minutes each time, 5-10 sessions for each course of BFT. During the therapy period, they were required to perform defecation force and relaxation training at home  two to three times every day, 20 minutes each time[27].

Round 2

Reviewer 2 Report

Dear Authors,

thank you for providing a carefully reviewed version of your paper. The requests have been successfully addressed. 

To be suitable for publication, there are some remaining issues which should be considered:

- In the results section, you need to present the matching between constipation groups according to the variables selected (age, gender, constipation course, and body mass index)

- Discussion (lines 333-334): the authors state that there is a significant difference in terms of QoL between groups. Indeed, they differ only for one of the items. So, this sentence should be rephrased in my opinion

- Please carry out language check (e.g., remove "more prevalence", "the colonoscopy", "and so on", "efficiency of BFT")  

- Conclusions:  "If an adult patient with FC shows RA, more attention should be paid to identify the severity of clinical manifestations and different treatments". The conclusion could be improved. Would you recommend a change in the diagnostic algorithm based on your findings? For example, upfront manometry?  

Reviewer 3 Report

The revised version is much better than the first version. However, the results of this study are quite different from the other studies. The prevalence of areflexia is very high, even though the authors used 50 mL rectal balloon distension. Other studies used 10-30 mL, and almost all participants had RAIR. So, I would suggest
1. add the reference regarding the prevalence of rectoanal areflexia from the Chinese or Asian population that is closer to your results. Or discuss why the prevalence of your study is different from the others.
2. It would be great if the authors could send me few more pictures of areflexia tracings of different participants.
